# The Role of Gratitude in a Positive Psychology Group Intervention Program Implemented for Undergraduate Engineering Students

**DOI:** 10.3390/bs13060460

**Published:** 2023-06-02

**Authors:** Antonios Kalamatianos, Kalliope Kounenou, Christos Pezirkianidis, Ntina Kourmousi

**Affiliations:** 1Department of Education, School of Education, University of Nicosia, Nicosia 2417, Cyprus; 2Student Counseling Center, School of Pedagogical & Technological Education, 15122 Maroussi, Greece; 3Department of Education, School of Pedagogical & Technological Education, 15122 Maroussi, Greece; kkounen@aspete.gr (K.K.); nkourmousi@aspete.gr (N.K.); 4Laboratory of Positive Psychology, Panteion University of Social & Political Sciences, Syggrou Ave. 136, 17671 Athens, Greece; christospez@hotmail.com

**Keywords:** positive psychology intervention, gratitude, happiness, optimism, undergraduate engineering students, counselling

## Abstract

Over the past decades, research on positive psychology for building strengths has proliferated. The present study aimed to explore the effect of gratitude in a 5-week positive psychology group program for undergraduate engineering students which included an intermediary 2-week gratitude intervention. In a mixed-design, 69 students from three engineering departments of the School of Pedagogical and Technological Education (ASPETE), assigned to the intervention (N = 34) and the control group (N = 35), with an average age of 21.52 years (*SD* = 4.63), were administered the Gratitude Questionnaire—six item form (GQ-6), the Modified Differential Emotions Scale (mDES), the Connor–Davidson Resilience Scale (CD-RISC), the Subjective Happiness Scale (SHS), and the Life Orientation Test—Revised (LOT-R). The condition experimental vs. control group was designated as the between-subjects factor, and time, that is, baseline vs. post intervention, was set as the within-subjects factor. Students who received the intervention reported significantly higher levels of gratitude. The increase in gratitude was due to the positive psychology group program. In addition, gratitude showed a significant effect on happiness and optimism, but failed to attain a significant impact on positive and negative emotions and resilience. Further research is needed to elucidate the effectiveness of positive psychology programs for undergraduate engineering students and the intervening cognitive processes.

## 1. Introduction

Positive psychology interventions (PPIs) are a relatively novel field consisting of “treatment methods or intentional activities that aim to cultivate positive feelings, behaviors, or cognitions” (p. 468) [1]. During the last 15 years, they have demonstrated various findings in promoting mental health in different settings, such as schools, workplaces, organizations, etc. [2,3]. According to the Broaden and Build theory [4], positive sentiments may change and reinforce thinking and behavior patterns and ultimately expand personal resources and promote physical and psychological well-being.

### 1.1. The Concept of Gratitude

Gratitude constitutes a positive psychology factor, which is displayed as a working mechanism of change in interventions using gratitude exercises that aim to promote well-being [5]. Gratitude has been deemed as a basic positive emotion that individuals experience in their daily lives [6] and a trait-like or dispositional attitude toward life [5]. The grateful emotion is considered as a “felt sense of wonder, thankfulness, and appreciation for life” (p. 460) [7]. It stems from the feeling one has when one personally benefits from another person’s intentional action, by a human or supernatural force [8], or the removal or absence of a negative factor or life condition [9]. As a consequence of feeling grateful, individuals tend to react positively and reciprocate an act of kindness [10].

More specifically, firstly, gratitude has been studied as a positive emotion theoretically [4,11,12] and empirically connected with other positive emotions [13]. A study [14], in accordance with the Broaden and Build theory [15], found that gratitude reduced the adverse effects of negative emotions. Secondly, as a character strength, gratitude is one of the emotional traits and indicates the individual’s tendency to recognize and respond emotionally by experiencing gratitude for acts of kindness that benefit him or her [14]. A person who exhibits high levels of gratitude may experience this trait more often, more intensely, and for a longer duration. In addition, individuals who are characterized as highly grateful have a greater range of recognition of gratitude sources. Finally, people who apply gratitude manage to connect every blessing or benefit they recognize in their lives to more than one source [14].

Three basic conditions must be met for individuals to develop high levels of gratitude as an element of their character, as stated by Watkins and his colleagues [16], the first being a strong, subjective sense of abundance, namely, a feeling that they have generally been “treated well”. Secondly, grateful people tend to appreciate the simple pleasures of everyday life and enjoy their fruits, without having to be given an extreme or intense trigger. Finally, a remarkably grateful person is characterized by an appreciation of others and their contribution to everyday life events and by the expression of this appreciation to others.

### 1.2. Gratitude and Well-Being

Numerous studies have analyzed gratitude’s relationship with social or physical well-being [17] and with positive emotions, life satisfaction, and subjective vitality [14,16] and have proven that it enhances the well-being of individuals [18,19].

A recent meta-analysis [20] has shown gratitude’s low to moderate effects on various psychological variables. Gratitude is associated with experiencing positive emotions, positive interpersonal relationships, meaning in life, and happiness [21,22,23]. Park et al. [24] pointed out that, out of the 24 character strengths they studied, only hope and zest had a stronger relationship with happiness than gratitude. Happiness may be viewed as the frequent experience of positive affect which leads to greater life satisfaction [25].

Furthermore, many experimental trials have examined the benefits of gratitude on mental health, and they have determined that gratitude is linked to positive psychological states, such as optimism and happiness, and plays a protective role against experiencing negative emotions [26,27]. Optimism may be seen as a person’s expectations of good outcomes [28]. People who experience gratitude have also been found to show elevated psychological resilience in the face of trauma-induced stress [29]. Resilience refers to one’s positive adjustment and effectiveness in recovering from negative experiences [30]. Gratitude has been empirically associated with resilience [27] and a fact that supports this notion is that high gratitude levels predict higher levels of well-being indices during adversities, such as the quarantine because of the COVID-19 pandemic [31,32]. Finally, higher levels of gratitude in adults and older adults have been associated with fewer psychological symptoms, such as anxiety and depression [16].

Alkozei et al. [18] proposed two causal models to explain the mechanisms through which gratitude contributes to individuals’ psychological flourishing based on the Broaden and Build theory [12]. According to the cognitive model, gratitude causes a cognitive expansion in the mind by helping it interpret negative or ambiguous events in a more positive way and focus more attention on the positive stimuli of each environment. These mechanisms, in turn, lead to the building of emotional, social, and physical resources, which are utilized when the individuals are faced with stressful events, leading to higher levels of well-being and consequently to a more frequent experience of gratitude acting as its feedback. The second model supports that experiencing gratitude expands the cognitive field of the mind by helping people find new ways to return the act of kindness to their benefactors, thus resulting in the creation of meaningful interpersonal connections and the building of positive relationships, which then lead to the availability of higher levels of social support. These mechanisms result in the achievement of higher levels of well-being, which afterward leads to a more frequent and intense experience of gratitude in everyday life.

### 1.3. Positive Psychology Interventions Focusing on Gratitude

Gratitude interventions include gratitude lists, in which the person counts their blessings [33,34,35], identifying sources of gratitude and simply reflecting on them or actively expressing gratitude, such as through a gratitude visit [23,34,35], and finally enhancing positive reframing [36]. The implementation of a blessing-counting exercise once a week increased participants’ gratitude and life satisfaction [33], while Seligman and his colleagues [23] experimentally applied a blessing-counting exercise to adults and the positive effects on their well-being were maintained for up to six months after the end of the intervention compared to the control group. Similar findings on the effectiveness of gratitude interventions in adolescent and child samples have been presented by other studies [37,38,39].

### 1.4. The Purpose of the Present Study

In the last five years, PPIs have been conducted in Greece, and their effectiveness in increasing well-being indices has been proven in several samples, such as children, emerging adults, the general population, and the elderly [40,41,42,43,44,45]. Although there has been great emphasis on academic issues with respect to university students and despite the plethora of papers on PPIs, there has been only limited published research pertaining to the use of positive psychology for the improvement of undergraduate engineering students’ quality of life in Greece [46,47]. Within higher educational settings, PPIs fitting non-psychology students need to be identified [48]. At the same time, students, as emerging adults, have to take on new responsibilities, experience adversities, and deal with a large number of challenges and difficulties [49]. Furthermore, bearing in mind, according to various meta-analyses [20,50], that gratitude has been empirically related to other positive psychology factors, that the effects of short-term positive psychology interventions, particularly of less than four sessions, may be small but significant, and that even interventions presenting small effect sizes can have a major impact on populations’ well-being, we decided to integrate a two-week intervention into a PPI group program of longer duration.

Moreover, other studies have reported high levels of stress in undergraduate engineering students [51] and the negative impact of the COVID-19 pandemic on their poor mental health conditions [52]. Consequently, an intervention, focusing on boosting the students’ resources, may be of great benefit to them [53], especially for engineering students that may need university counseling services to assist them in confronting their challenges with a positive mindset and an anti-deficit perspective [54]. Keeping in consideration that engineering may appeal to students who are more adept at systemizing and less adept at empathizing [55] and who value scientific tasks more than altruistic tasks [56] and that even brief intervention may have beneficial results in engineering students [57], the requirement for psychological intervention for this specific population becomes more apparent.

Based on the aforementioned information, the Counseling Center of the School of Pedagogical and Technological Education (ASPETE) aims to explore the effect of gratitude in a positive psychology group intervention implemented for undergraduate engineering students and altering the viewpoint away from student shortcomings and toward the acknowledgment of their mental and emotional qualities [58], applying a multicomponent PPI approach, including several empirically tested positive psychology activities, since it is more effective than engaging in one project [1]. The current exploration of gratitude in an academic setting was part of a larger study, focused on positive emotions, and part of a psychoeducational program applied to undergraduate engineering students [46].

Overall, the purpose of this study was to examine the role of gratitude in a group PPI program implemented for undergraduate engineering students. To the best of our knowledge, the effect of gratitude in terms of a positive psychology, psycho-educational group program has not been examined so far in Greek undergraduate engineering students. In particular, we expected that (a) the students in the experimental group will score significantly higher in gratitude between baseline and post test, (b) the students in the control group will report no significant change in gratitude between baseline and post test, and (c) the students in the experimental group will report significantly increased levels of gratitude compared to students in the control group at post test. In addition, we will explore the effect of gratitude on the positive psychology variables in the experimental group.

## 2. Materials and Methods

### 2.1. Participants

A convenience sample of 164 students from three engineering departments of ASPETE, 75 from civil and mechanical engineering educators, for the experimental group, and 89 from electrical engineering educators, for the control group, were recruited. Out of the students who took part at the beginning, 41 individuals from the experimental group were excluded because they did not fully attend the program, while 54 individuals from the control team were lost in the post-test period. Hence, a total of 69 people, 52 men and 17 women, formed the final sample of the study. The experimental group and the control group had an average age of 22.09 years (*SD* = 6.52) and 20.97 years (*SD* = 0.99), respectively. In terms of gender, the experimental group consisted of 22 males (64.7%) and 12 females (35.3%), and the control group contained 30 males (85.7%) and 5 females (14.3%). The Chi-square test displayed a significant difference in the weak effect size (Cramer’s V = 0.24) in the two groups (χ^2^ = 4.10, *p* < 0.05).

### 2.2. Design and Procedure

The study was carried out in accordance with the Declaration of Helsinki’s ethical principles. It also abided by the operating regulations of the student counseling center that were approved by the research and management committee of ASPETE (No. 48, 19-12-2018, item 5.14) and by the head committee of ASPETE (the act of the meeting of the management committee, No. 45, 19-12-2018, item 3.1) and regulate all issues related to the center’s operation, such as confidentiality, privacy, security, and data protection.

In this study, a mixed measures design was employed. More specifically, the within-participants factor was time, that is, baseline vs. post-intervention, and the between-subjects variable was the condition experimental vs. control group. The students were invited to participate in a study that was announced as a psycho-educational program concerning the use of positive psychology for the improvement of one’s mental health. Personal codes for every participant were used to assure confidentiality and anonymity. The individuals did not receive any reward for their participation.

The psychoeducational program was carried out in the university setting, during students’ three specific classes, on the same day every week for each department by the psychologists working in the student counseling center. Initially, students from both groups, after signing a consent form, completed the following questionnaires: The Differential Emotions Scale-modified (mDES), the Connor–Davidson Resilience Scale (CD-RISC), the Subjective Happiness Scale (SHS), and the Life Orientation Test—Revised (LOT-R), which were re-administered at the end of the program. The group intervention contained five 1.5–2 h psychoeducational seminars and was offered over 5 weeks to the students of the experimental group. We drew techniques from different positive psychology intervention programs offered to young adults, mostly, the 8-week positive psychotherapy group by Parks and Seligman [59], as well as *The How of Happiness* [60], and the acts of kindness interventions [25] to increase the possibility of a prolific outcome [61].

More specifically, the group program included short lectures that were delivered to the students so as to introduce them to new material in each session. We presented information about positive psychology and its main topics, as well as the general aim of each exercise. We also stressed how each technique can help students. The program comprised various exercises, such as mindful attention, savoring, the three good things, using one’s strengths, kindness, life summary, and personal motto, suiting the students’ needs.

In the third session, the students of the experimental group were administered the Gratitude Questionnaire—six item form (GQ-6). Then, in the course of the session they were provided with a brief lecture pertaining to gratitude, the theoretical framework for it, and its benefits [33], and they were handed a worksheet with the gratitude journal as homework, where the participants were asked to list things daily that they would feel grateful for in the following week. Subsequently, in the fourth session, they completed a gratitude letter, where they were invited to write down how they profited from the actions of a benefactor and thank that person for the benefit they received. The rationale behind the selection of exercises was to incorporate two types of methods, self-reflective, such as the journal, where they could express themselves, and interactive practice, which would probably give them the opportunity to, more actively, put forward a token of their appreciation to a person of their choice. Next, in the fourth session, they re-administered the gratitude measure. Simultaneously, the students of the control group were administered the GQ-6 at the beginning of their class in the third week and at the end of their class in the fourth week.

At the end of the group intervention, we gave an overview of the program, and the students supplied their feedback. The participants in the control group received no intervention (see Appendix A).

### 2.3. Measures

An improvised basic demographic questionnaire included general background information on gender, age, and faculty.

Grateful disposition was measured by the Gratitude Questionnaire—six item form (GQ-6) [14]. There are six items on the scale (“I have so much in life to be thankful for”) and ratings are made on a 7-point Likert scale (1 = “strongly disagree”, 7 = “strongly agree”). The degree of the usual experience of gratitude, its frequency, and the people to whom someone is grateful to are explored by the GQ-6 [62]. The GQ-6 has demonstrated good internal reliability, with alphas ranging between 0.82 and 0.87, and it has been found to be positively related to prosociality and measures of positive affect and well-being, for instance, satisfaction with life, subjective happiness, vitality, optimism, and hope and negatively related to depression, anxiety, materialism, and envy [35,62,63]. The Cronbach’s alpha of the GQ-6 in this study was 0.81.

Emotions were assessed by the Modified Differential Emotions Scale (mDES) which was created by Izard [64] and modified by Fredrickson [65]. It measures the specific distinct, positive, and negative emotions that the respondents had experienced within the last 2 weeks. The test contains 21 questions, and the answers range from 1 to 5. The Greek standardization displayed satisfactory reliability [66]. The Cronbach’s alphas of the positive emotions and the negative emotions subscales in the present study were 0.82 and 0.84, respectively.

Resilience was assessed by the Connor–Davidson Resilience Scale (CD-RISC) [67]. CD-RISC measures one’s ability to overcome adversities and, particularly, tenacity and competence, trusting in one’s instincts and tolerance of negative affect, acceptance of change and security within relationships, control, and spirituality. It includes 25 items, and responses are given on a four-point Likert scale. It has exhibited good psychometric properties. The Cronbach’s alpha of the CD-RISC in this study was 0.86.

Happiness was measured by the Subjective Happiness Scale (SHS) [68]. Participants answer 4 statements on a 7-point Likert scale. The Greek standardization [69] has shown satisfactory reliability indexes. As for criterion validity, the SHS was significantly negatively correlated to negative emotions, stress, anxiety, and depression and positively correlated to life satisfaction, psychological resilience, and hope. A Cronbach’s alpha of 0.84 was obtained in this research.

The Life Orientation Test—Revised (LOT-R) was used to identify one’s dispositional level of optimism. LOT-R, developed by Scheier and Carver [28] and modified by Scheier et al. [70], contains 10 items rated on a 5-grade Likert scale. The respondents state their expectations, in general, regarding future outcomes. The LOT-R has manifested adequate psychometric properties, such as coefficient alpha reliability, which was found to be 0.78, and convergent and discriminant validity. The internal consistency coefficient for the Greek adaptation of the total scale was 0.71 [71]. A Cronbach’s alpha of 0.72 was obtained in this paper.

### 2.4. Statistical Analysis

There was no deviation from linearity found in the initial test that we performed with the scatter plots. We also used histograms, Q-Q plots, the goodness of fit tests, e.g., the Kolmogorov–Smirnov and the Shapiro–Wilk tests, and skewness and kurtosis values to check for normality. We found no statistically significant violations [72,73]. In the present study, we focused on the role of gratitude in a group intervention that contains several positive psychology measures, as the differences between the two groups in the measures administered in the first and the fifth week have been elucidated in another paper [46]. Consequently, independent *t*-tests at baseline and post intervention were estimated to compare the two groups in gratitude and a paired-sample *t*-test was conducted to compare gratitude in each group, at two time points, baseline and post test. In order to examine the effect of the intervention, we carried out a mixed ANOVA with the experimental condition as the between-participants factor and time (baseline vs. post-test) as the within-subjects variable. Since we found a statistically significant change in gratitude only in the experimental group, we further investigated gratitude’s connection with the other positive psychology variables and calculated correlations between gratitude and positive and negative emotions, resilience, subjective happiness, and optimism. Finally, a multivariate analysis was conducted to determine the effect of gratitude on the other measured variables in the intervention group. We used SPSS version 25 (IBM, Armonk, NY, USA).

## 3. Results

The independent variable was the two-level (experiment vs. control) condition, and the dependent variable was gratitude. The participants who received the positive psychology intervention (*M* = 5.09, *SD* = 0.88) compared to the participants in the control group (*M* = 4.98, *SD* = 0.78) demonstrated non-significantly higher pre-PPI scores, *t*(67)= 0.54, ns, and significantly higher scores (*M* =5.53, *SD* = 0.98; *M* = 4.88, *SD* = 0.85, respectively) *t*(67)= 2.98, *p* = 0.004, after the intervention.

Regarding the within-subject differences, we used a dependent samples *t*-test. No statistically significant results were drawn from the control group, *t*(34) = 1.14, ns. On the contrary, in the experimental group, the results from the pre-design and post-design measures indicated that the PPI resulted in a statistically significant improvement in gratitude levels, *t*(33) = −3.39, *p* = 0.002 (see Appendix A).

Finally, a mixed ANOVA demonstrated a statistically significant interaction between condition and time for gratitude, *F*(1, 67) = 11.889, *p* < 0.001, *η_p_*^2^ = 0.15 (see Appendix A).

Hence, since we did not discover statistically significant changes in the control group at the end of the fifth week in comparison with the first week, we focused only on the post-design scores of the experimental group. The majority of the correlations between the gratitude assessed at the end of the fourth session, that is, after the gratitude intervention, and the post-test assessments of all the other measures, at the end of the fifth session, were positive and statistically significant. In particular, the strongest correlation was found between post-test gratitude and subjective happiness, *r* = 0.69, *p* < 0.001, while gratitude was found to be also positively related to resilience, *r* = 0.63, *p* < 0.001, positive emotions, *r* = 0.60, *p* < 0.001, and optimism, *r* = 0.51, *p* = 0.002, while negatively related, yet non-significantly, to negative emotions, *r* = −0.28, *p* = 0.106, ns.

Multivariate analysis was performed only in the intervention group and revealed that the effect of gratitude on subjective happiness *F*(1, 18) = 3.45, *p* = 0.009, *η_p_*^2^ = 0.81, and optimism *F*(1, 18) = 2.47, *p* = 0.042, *η_p_*^2^ = 0.75, was statistically significant, whereas gratitude’s effect on resilience, *F*(1, 18) = 2.16, ns, *η_p_*^2^ = 0.72, positive emotions, *F*(1, 18) = 1.46, ns, *η_p_*^2^ = 0.64, and negative emotions, *F*(1, 18) = 1.16, ns, *η_p_*^2^ = 0.58, was non-significant (see Appendix A).

## 4. Discussion

The main goal of the current study was to examine the effect of gratitude on other human resource strengths that were developed by positive psychology, such as positive and negative emotions, happiness, optimism, and resilience, in terms of a group psychoeducational intervention for undergraduate engineering students. We analyzed the results of a 2-week gratitude intervention, integrated into a 5-week program that has already been shown to lead to significantly higher levels of positive emotions, subjective happiness, and optimism in the intervention group compared with the control one [46], on gratitude scores in comparison with a control condition. The main finding of the research showed that the intervention managed to significantly increase the grateful emotion in the experimental group only, whereas there was no change in the control group. Subsequently, in the intervention group, the elevation of gratitude was due to the implementation of the psychoeducational program. It looks like the positive psychology psychoeducational-program-embedded gratitude exercises offered individuals a method of identifying and reflecting on the things they were grateful for, and within two weeks they experienced a significant boost in the grateful sentiment. Finally, gratitude exerted significant effects on happiness and optimism but failed to connect with positive emotions and resilience. Thus, it seemed that counting their gratitude instances significantly bolstered happiness and optimism.

These findings are in accordance with previous meta-analysis results that have shown the improvement of happiness by reason of the expression of gratitude for significant periods of time [25]. Likewise, Seligman et al. [23], testing an online self-selected community sample, pointed out that happiness increased over a 6-month period when the participants considered three good things that happened to them each day over 1 week and wrote and delivered a letter of gratitude. Other previous studies support gratitude’s positive relationship with happiness [74,75,76]. Two additional lines of research have also proven the link between gratitude journaling and optimism compared to active control and no-treatment groups [77,78]. Contrary to other research [79], gratitude did not have an intensified impact on resilience. More than two weeks of gratitude practice may have been necessary to help students better manage any hardships they encountered [80,81]. Finally, in contrast to our results, McCanlies et al. [27] reported that gratitude was positively related to positive emotions. Seemingly, the two-week writing interventions were not enough to elicit statistically significant changes in the total score of the positive or negative emotions of the students. Additional exercise time is probably needed in order for gratitude feelings to cause a cognitive expansion that will help individuals interpret negative events in a more positive way and lead to the building of personal and interpersonal resources [18,82].

The lack of consistent findings regarding the effects of gratitude on the positive emotions of young adults, students, and clinical samples is due to various reasons, such as the gratitude theme or the time frame in which a gratitude intervention is deployed [83]. Lyubomirsky [60] reported that PPIs, requiring stable gratitude practice, exhibit a positive correlation with greater happiness, more energy, hope toward the future, and positive emotions. One of the studies, testing the effectiveness of gratitude interventions, asked students to reflect on things that had happened to them over the summer that they felt grateful for. Students in this group reported higher levels of gratitude and lower levels of experiencing negative emotions compared to students in the control group [16]. Additionally, another major study on the effectiveness of gratitude interventions concluded that implementing a blessing-counting exercise once a week increased individuals’ well-being, specifically the experience of positive emotions, gratitude, and life satisfaction, by up to three months after the intervention in both healthy populations and populations suffering from neuromuscular diseases [33].

Our study has several limitations. Firstly, the intervention was implemented on a small number of students, and the participants in the two groups were not matched. Secondly, we did not translate the inner action into outward action, that is, we did not include behavioral expressions of gratitude, for example, a gratitude visit, in the assigned exercises. Moreover, the gratitude intervention only had a two-week duration and was of an exploratory nature. Our main focal point was only the impact of the gratitude exercises, and not the other intervention program exercises, on the measures used in this study. Lastly, the subjects were not randomly assigned into the two groups, and, thus, we could not control for possible confounders outside of the intervention environment. Long-term group psychoeducational programs may offer students the opportunity to convert the positive activities they practice into habits.

Notwithstanding the shortcomings, this paper has advantages as well. Despite the abundance of studies on positive psychology, to the best of our knowledge, relatively little research has been conducted and published on the role of gratitude in positive psychology group interventions at universities, especially undergraduate engineering students [48], in Greece regarding the amelioration of the quality of their lives. In addition, the inclusion of a control group ensures the internal validity of the study. With respect to future directions, randomized controlled trials with greater and more representative samples could be useful in discovering long-lasting results. Thus, in this regard, long-term research with follow-up sessions, related to the grateful strategies the students may choose in order to extract benefits from adverse experiences, such as exam failure, etc., stands out as a priority. Moreover, except for cognitive processes, students’ academic achievement also involves emotional states [84], especially positive emotional functioning, that should be explored extensively. Apart from that, positive psychology interventions embody a set of strategies that concentrate on ameliorating well-being and positive cognitions and emotions [85]. Hence, an overarching goal of positive psychology intervention research especially in undergraduate engineering students, namely, of a field that by and large remains so far inadequately explored, may be to break down learning patterns and emphasize students’ strengths to increase their performance. Last, future research could choose different tools, related to positive psychology, and compare Greek engineering educators with students of other faculties and countries.

## 5. Conclusions

In summary, this paper emphasized the significant role of gratitude in factors that help undergraduate engineering students evaluate life in a positive way. Gratitude may prove to be useful for them, since by maintaining a grateful focus, they may overcome various obstacles that they confront during their studies. They seem to respond to others’ beneficence and acknowledge they are recipients of benevolent behavior. Thus, psychoeducational and counseling interventions that aim to foster undergraduate engineering students’ gratitude can have a positive impact on their well-being.

## Data Availability

The raw data supporting the conclusions of this article will be made available by the authors, without undue reservation.

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
