# Peer review of "The Role of Gratitude in a Positive Psychology Group Intervention Program Implemented for Undergraduate Engineering Students"

_behavsci, 2023, doi:10.3390/bs13060460_

Round 1

Reviewer 1 Report

Dear authors,

thank you for your manuscript. I was very interested to know if the intervention could be successful. Regretfully, it is unclear what the exact method was. Which time frame was researched? The whole intervention course or only the two weeks of the gratitude intervention? Were all tests administered to both groups? Were all tests done at the start of the gratitude session and again two weeks later after the second session? Due to this questions it is not possible to follow your results or the discussion. There are a lot of open questions I also commented on in the manuscript. Please clarify these issues to allow an understanding of your work.

Best wishes

Author Response

Dear reviewer,

thank you very much for your constructive comments that helped us enhance the quality of our manuscript.

Attached you can find our answers to your comments.

Reviewer 2 Report

A well-written paper. The weaknesses which are important are noted by the authors. There is the problem that the two groups come from different classes, so that is an interesting conflict. This is clearly a preliminary paper, noting the small 'n' and short time duration of the experiment. However, it does provide insight into a more in-depth study to be conducted in the future and as such deserves publication.

Minor English language changes include: line 38, change "he/she" to "they"; line 66 change "to" to with; line 161, change "sex" to "gender". Overall language quality is excellent and the text easy to read. I raise a note of caution to avoid modern-speak with respect to pronouns. There are many perfectly-acceptable pronouns without diverging into political write-speech. Stick to the 'old' English for journal publications as it better fits the audience who will read the paper.

Author Response

(The authors gave the same response as above.)

Reviewer 3 Report

1. Very interesting study, but why only engineering students?

2. A very complex study to measure a very subjective sensation, but other scales can be used or a more adequate scale can be created.

3. What is your concept of happiness?

4. It was good to compare studies with students from other countries or remote regions.

5. It never hurts to double-check references in the text and revise the native language.

6. Good initiative, especially in the now post-pandemic period.

Author Response

(The authors gave the same response as above.)

Round 2

Reviewer 1 Report

Dear authors,

thank you very much for the revised version of the manuscript. My understanding of the methods was much improved due to your changes. I noticed one point. You did not look at the differences between the measures in the first and last week. Should it not be probable, that they too changed due to the intervention? I think it would further strengthen your discussion if you included the analyses containing this.

Best regards

Author Response

Dear Sir,

Thank you very much for your notice. Actually, we analyzed the differences between the measures in the first and last week, but this analysis is part of a prior article, already published. We have mentioned in our paper that this study was part of a research that included the questionnaires used in the present one. So, we added text in the Statistical analysis and the Discussion, containing information with regard to the results of the aforementioned analysis.